# Meta-analysis of efficacy and safety of sustained release oxycodone hydrochloride rectal administration for moderate to severe pain

Xian Bing Hou[1]*, Dan Dan Chen[2‡], Tong Fei Cheng[3], Dan Wang[1], Xiao Jun Dai[1], Yao Wang[1], Bi Xian Cui[1], Yuan Yuan Wang[1], Hui Xu[1], Hong Zhou Chen[1]

1 Department of Oncology, Fenghua District Hospital of Traditional Chinese Medicine, Ningbo, Zhejiang Province, China, 2 Department of Acupuncture, Fenghua District Hospital of Traditional Chinese Medicine, Ningbo, Zhejiang Province, China, 3 Department of Nursing, Fenghua District Hospital of Traditional Chinese Medicine, Ningbo, Zhejiang Province, China

‡ DDC contributed equally listed as co-first author.
* houxianbing007@126.com

**Data Availability Statement:** All relevant data are within the paper.

**Funding:** This work was supported by the Science and Technology Project of Fenghua Science and

## Abstract

### Objective

This study aims to evaluate the efficacy and safety of oxycodone hydrochloride (OxyContin) rectal administration in cancer pain patients. This is geared towards providing the research evidence for a novel route of OxyContin administration.

### Methods

Relevant randomized controlled trials (RCTs) were searched in electronic databases, including PubMed, Cochrane Library, Web of Science, EMBASE, China National Knowledge Infrastructure (CNKI), Chinese Scientific Journal Database (VIP database), Wanfang Data Knowledge Service Platform, and Chinese Biomedical Literature Database (CBM). Moreover, unpublished academic data were obtained by contacting the colleague, professor, or Institute of Traditional Chinese Medicine. The RCTs of transrectal Oxycodone administration of sustained-release tablets for moderate and severe pain patients were searched in the databases from inception to December 2020.

### Results

According to the inclusion criteria, a total of 8 RCTs were included, with a total of 648 patients. Meta analysis results showed that there was no statistically significant difference in the efficacy of moderate to severe pain control between the rectal administration group and the oral administration group (RR = 1.04, 95%CI: 0.99–1.10, p = 0.13>0.05). At the same time, the incidence of adverse reactions in the rectal administration group was low. In terms of constipation, the rectal administration group was less than the oral administration group, with a statistically significant difference (RR = 0.43, 95%CI: 0.31–0.58, p< 0.00001). In terms of nausea and vomiting, the rectal administration group was less than the oral

Technology Bureau (NO: 20186515). The funders had no role in study design, data collection and analysis, decision to publish, or preparation of the manuscript. There are no authors received a salary from any of our funders.

**Competing interests:** The authors have declared that no competing interests exist.

**Abbreviations:** CBM, Chinese Biomedical Literature Database; CNKI, China National Knowledge Infrastructure; VIP database, Chinese Scientific Journal Database; CINV, chemotherapy induced nausea and vomiting; RCTs, relevant randomized controlled trials; RR, relative risk; 95%, CI 95% confidence interval; NRS, numerical rating scale; EG, experimental group; CG, control group.

administration group, and the difference was statistically significant(RR = 0.30, 95%CI: 0.21–0.42, p<0.00001). In terms of sleepiness, there was no significant difference between the two groups(RR = 0.54, 95%CI: 0.26–1.15, p = 0.11>0.05). In terms of dizziness, there was no statistically significant difference between the two groups (RR = 0.43, 95%CI:0.27–0.68, p = 0.31>0.05). In terms of dyuria, there was no statistically significant difference between the two groups (RR = 0.37, 95%CI: 0.02–7.02, p = 0.51>0.05). In terms of KPS scores there was no significant difference was noted between the rectal and oral administration groups (RR = 1.04, 95%CI: 0.89–1.21, p = 0.63>0.05).

## Conclusion

In summary, we found no significant differences in efficacy between rectal administration of OxyContin and oral administration. Thus, rectal administration should be considered in managing cancer pain among patients with difficulty in oral OxyContin administration.

## PROSPERO registration number

CRD42021209660.

## Introduction

Clinical pain treatment adopts a three-step treatment plan by the World Health Organization (WHO). Using the NRS score as a standard, the pain was divided into mild pain (1-3points), moderate pain (4-6points), severe pain (7-10points) [1, 2]. Moderate to severe pain generally requires opioids for treatment. OxyContin is one of the commonly used opioids whose conventional administration route is oral. However, its clinical application has encountered challenges. The administration method of analgesic therapy requires non-invasiveness, and oral administration is the preferred treatment route. Nonetheless, a few patients cannot be administered orally because of severe nausea, vomiting, difficulty eating, gastrointestinal reactions, intestinal obstruction, etc. Notably, injection therapy causes iatrogenic pain and patient dependence on the hospital; it also affects the psychology of the patient. In addition, rectal administration is a non-invasive method of administration; it has few requirements to a certain extent and is suitable for nearly all patients experiencing pain. Although oral administration is the most convenient route, it cannot be applied as the first choice in many cases. Therefore, rectal administration is significant to patients with difficulty in oral administration. At present, many clinical studies utilize rectal administration to resolve the above problems, with effective treatment outcomes. Nevertheless, these studies used a small sample size, hence the findings are not convincing. Thus, it is necessary to systematically evaluate the efficacy and safety of rectal OxyContin administration in the management of moderate to severe pain. This work conducted a meta-analysis of the efficacy and adverse effects of rectal OxyContin administration, geared towards providing evidence-based information for a new route of OxyContin administration.

## Materials and methods

This systematic review protocol is registered on PROSPERO (https://www.crd.york.ac.uk/prospero/display_record.php?ID=CRD42021209660). The registration number is CRD42021209660. This is a literature-based study thus ethical approval was unnecessary.

## Literature resources

**Inclusion criteria.** The inclusion criteria included; clinical trials analyzing the transrectal administration of OxyContin for moderate to severe pain management; a randomized controlled trial; The control group treated with oxycodone sustained-release tablets.

**Exclusion criteria.** Clinical trials excluded included; Case reports; animal experiments; basic research; personal experience and review literature; incomplete data; Duplicate literature; Baseline conditions not evaluated.

**Data extraction.** All the records were fed into the EndNote X8 software after the electronic search stage. Data were independently extracted by 2 researchers (CTF and WD) using a predefined data extraction form. Two researchers independently screened titles and abstracts to establish the trials to be excluded. The full text was examined if necessary. The following aspects were considered: general information (year of publication, author's details, etc.), participations, inventions, comparisons, outcomes, adverse events, and other information. Two authors resolved disagreement by discussion, and a third author (XH) arbitrated if there were further disagreements.

**Intervening measure.** The experimental group was administered with OxyContin (Oxycodone, Beijing Mengdi Pharmaceutical Co. LTD.) by rectum, while the control group was administered with oxycodone sustained-release tablets by mouth, with the same initial dose.

## Search strategy and quality assessment

**Document retrieval.** Two reviewers (CDD and HXB) independently searched the studies in electronic databases based on the systematic review. Two retrieval personnel with "oxycodone, oxycontin, dosing, and anal rectum and anus" and "pain, cancer pain, pain, cancer pain", "randomized, RCT" as keywords, computer retrieving CNKI, VIP, CBM, Pubmed, Embase, the Cochrane library, retrieval of oxycodone Zyban rectum for drug efficacy and safety of the treatment of pain. The retrieval period was from the inception of the database to December 2020, and the relevant original data of the literature was extracted (Table 1).

**Quality assessment.** Quality assessment was conducted using the Cochrane collaboration's risk of bias tool. Two authors (CBX and WYY) estimated the domain risk of bias as follows: Sequence generation of randomized, allocation concealment, blinding of participants, personnel and outcome assessment, incomplete outcome data and selective outcome report, and other sources of bias. Any disagreements were resolved by a third author (CHZ).

## Statistic analysis

**Outcome indicator.** *Primary outcomes.* Regarding the effective rate of pain control, clinical efficacy was considered the primary outcome indicator. The therapeutic efficacy standard was divided into 4 grades based on the WHO pain treatment remission (PAR) [3], i.e., Complete remission (CR): No pain after treatment; Partial relief (PR): the pain was significantly relieved compared to that before treatment, sleep was undisturbed, and normal life could be achieved; Mild relief (MR): The pain was less than before treatment, but the pain was still felt, and the sleep is disturbed; No relief (NR): No change in pain compared to before treatment. Effective rate = (CR+PR) number of cases/total cases × 100%.

*Secondary outcomes.* The secondary outcomes were any adverse events including constipation, nausea, vomiting and dizziness, and quality of life, which were measured using the Karnofsky scale.

**Meta-analysis.** The RevMan5.3 software was used for statistical analyses. The evaluation index of the study was the data of dichotomous variables, including the effective rate of pain control and incidence of adverse reactions. The risk ratio (RR) was used as the analysis

**Table 1. Pubmed search strategy.**

| Search number | Search Details | Results |
|---|---|---|
| 9 | (("Oxycodone"[MeSH Terms] OR (("Oxycone"[Title/Abstract] OR "oxycodone hydrochloride"[Title/Abstract]) OR "Oxycontin"[Title/Abstract])) AND ("administration, rectal"[MeSH Terms] OR (((("Drug"[All Fields] AND "administration anal"[Title/Abstract]) OR "drug administration rectal"[Title/Abstract]) OR ("Anal"[All Fields] AND "drug administration"[Title/Abstract])) OR "rectal administrations"[Title/Abstract]))) AND "Randomized Controlled Trial"[Publication Type] | 3 |
| 8 | "Randomized Controlled Trial"[Publication Type] | 512,552 |
| 7 | ("Oxycodone"[MeSH Terms] OR (("Oxycone"[Title/Abstract] OR "oxycodone hydrochloride"[Title/Abstract]) OR "Oxycontin"[Title/Abstract])) AND ("administration, rectal"[MeSH Terms] OR (((("Drug"[All Fields] AND "administration anal"[Title/Abstract]) OR "drug administration rectal"[Title/Abstract]) OR ("Anal"[All Fields] AND "drug administration"[Title/Abstract])) OR "rectal administrations"[Title/Abstract])) | 9 |
| 6 | "administration, rectal"[MeSH Terms] OR (((("Drug"[All Fields] AND "administration anal"[Title/Abstract]) OR "drug administration rectal"[Title/Abstract]) OR ("Anal"[All Fields] AND "drug administration"[Title/Abstract])) OR "rectal administrations"[Title/Abstract]) | 3,127 |
| 5 | "Oxycodone"[MeSH Terms] OR "Oxycone"[Title/Abstract] OR "oxycodone hydrochloride"[Title/Abstract] OR "Oxycontin"[Title/Abstract] | 2,419 |
| 4 | (((("Drug"[All Fields] AND "administration anal"[Title/Abstract]) OR "drug administration rectal"[Title/Abstract]) OR ("Anal"[All Fields] AND "drug administration"[Title/Abstract])) OR "rectal administrations"[Title/Abstract] | 610 |
| 3 | "administration, rectal"[MeSH Terms] | 2,531 |
| 2 | "Oxycone"[Title/Abstract] OR "oxycodone hydrochloride"[Title/Abstract] OR "Oxycontin"[Title/Abstract] | 382 |
| 1 | "Oxycodone"[MeSH Terms] | 2,282 |

statistic, and the 95% confidence interval (CI) was calculated. RR was analyzed by the Z test. $P < 0.05$ was considered a statistically significant difference in evaluation indices between the two groups.

**Heterogeneity test.** Based on the test level of $\alpha = 0.05$, there was inter-study heterogeneity if $P < 0.05$. At the same time, heterogeneity of $I^2$ was quantitatively analyzed, $I^2 \geq 50\%$ and the inter-study heterogeneity was large. In the absence of heterogeneity, a fixed-effect model was selected for combined effect analysis, while a random effect model was used.

**Assessment of reporting bias.** Funnel plots were used to assess the presence of publication bias. Reporting bias was also assessed using Egger's test and Begger's analysis.

**Sensitivity analysis.** With the availability of sufficient trials, sensitivity analysis was performed by sequentially eliding each trial to examine the robustness of the final results.

**Strategy for data synthesis.** The RevMan V.5.3 software was used for data synthesis. Dichotomous data were expressed in RR and continuous data in mean difference (MD). The fixed-effect model was used if $I^2 < 50\%$ or $I^2 > 75\%$, $I^2 < 50\%$ had low heterogeneity, whereas those with $I^2 > 75\%$ had high heterogeneity. A subgroup analysis or a sensitivity analysis was performed when $I^2 > 75\%$.

# Result

## Document retrieval process

A total of 175 related literature were retrieved (CNKI = 28; VIP = 36; Wanfang = 76; CBM = 26; Pubmed = 3; Embase = 2; Cochrane Library = 4); 86 repeated references were excluded; 2 references on exclusion system evaluation; 41 references with unrelated research contents, 27 with

different intervention measures; 2 with non-randomized controlled trials; 6 with non-oxycontin treatment were excluded, and 8 were included in the meta-analysis [3–10]. (Fig 1).

## Basic information of literature

The major characteristics of all included studies were showed on Tables 2 and 3. Our analysis include the effective rate of pain control and any adverse events including nausea, vomiting and dizziness, constipation, dysuria and lethargy.

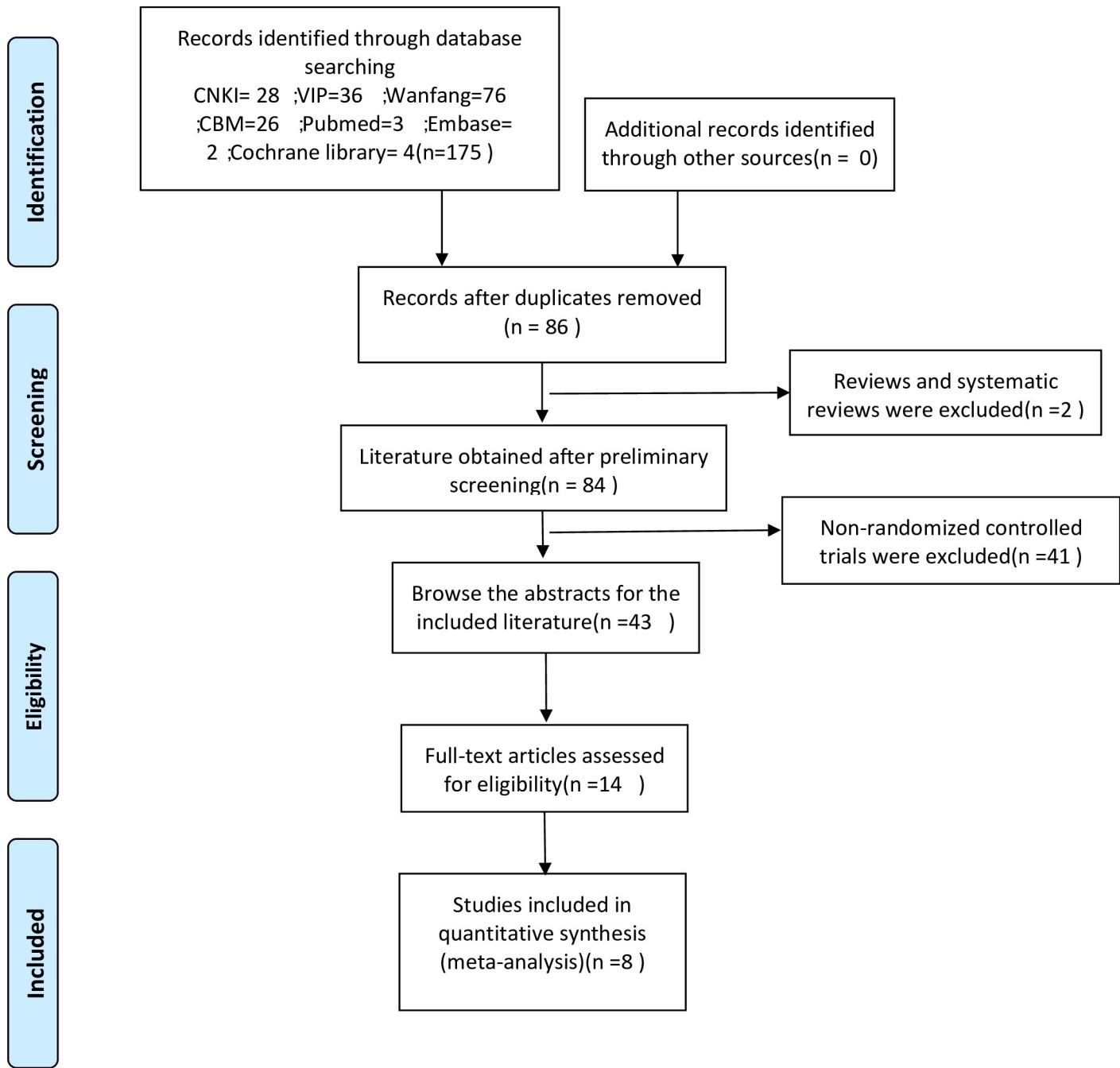

**Fig 1. Flow diagram of the literature search.**

## Literature quality evaluation

The bias risk assessment of the included studies were showed on Figs 2 and 3. Fig 2 shows the proportion of studies assessed as low, high or unclear risk of bias for each risk of bias indicator. Fig 3 shows the risk of bias indicators for individual studies. The method of randomization was described by all eight trials which have no describe allocation concealment methods.

## Meta analysis results

**Bias test.**    A funnel plot (Fig 4) was drawn to investigate a publication bias. $P = 0.553 > 0.05$ based on Egger linear regression method and $P = 0.548 > 0.05$ based on the Begg rank correlation test suggested significant publication bias in the included literature.

**Effective rate of pain control.**    All 8 literature reported an effective rate of pain control. After the heterogeneity test, $I^2 = 71\% > 50\%$, and $p = 0.001 < 0.1$, indicating that the heterogeneity among the selected pieces of literature was statistically significant, and a heterogeneity search was required. Sensitivity analysis was conducted on the 8 literature, and we found that du yajuan 2020 had a significant effect on heterogeneity. After removing this study, heterogeneity test analysis was repeated, and the results indicated that the remaining 7 [4–10] literature had no heterogeneity ($I^2 = 41\%$, $p = 0.12 > 0.1$). After the exclusion, meta-analysis was conducted with fixed effects. The RR of the 7 studies was 1.04 with the 95% confidence interval of 0.99–1.10, which was statistically significant ($z = 1.53$, $p = 0.13 > 0.05$, Fig 5); this suggests no statistical difference between the efficacy of oxycodone sustained-release oxycodone tablets (oxycontin) for rectal administration of moderate to severe pain and oral administration.

**Security analysis.**    Adverse reactions were reported in all the 8 literature, primarily nausea and vomiting, constipation, drowsiness, and dizziness. The incidence of nausea, vomiting, and constipation was lower in rectal administration group than that of oral oxycodone sustained-release tablets, and the difference was statistically significant ($p < 0.05$). No significant difference was noted between the two groups in drowsiness, dizziness, and dyspnea. Besides the above five major adverse reactions, Wu Reported psychiatric symptoms, and the rectal administration group was less than the oral administration group. Du Yajuan reported skin itching, rectal administration of excess oral medication in the group. In terms of compliance, the rectal administration group was superior to the oral administration group, with a statistically significant difference. In addition, no adverse reactions including addiction, hypotension, and respiratory depression appeared in the statistics.

*Constipation*. Seven studies [3–6, 8–10] reported the incidence of constipation with no heterogeneity between studies. The fixed-effect model was used for analysis, and the difference was statistically significant (RR = 0.43, 95%CI: 0.31–0.58, p< 0.00001, Fig 6).

*Nausea and vomit*. Seven studies [3–6, 8–10] reported the incidence of nausea and vomiting with no heterogeneity between studies. The fixed-effect model was used for analysis, and the difference was statistically significant (RR = 0.30, 95%CI: 0.21–0.42, p < 0.00001, Fig 7).

*Lethargy*. Five studies [6, 8–11] reported the incidence of narcolepsy, with no heterogeneity between studies. The fixed-effect model was used for analysis, and we found no significant difference between the rectal and oral administration groups (RR = 0.54, 95%CI: 0.26–1.15, p = 0.11, Fig 8).

*Dizziness*. Four studies [4, 9–11] reported the incidence of dizziness, with inter-study heterogeneity (p = 0.003, $I^2 = 79\%$), and the random-effect model was used for analysis. No statistically significant difference was noted between the two groups (RR = 0.43, 95%CI: 0.27–0.68, p = 0.31 > 0.05, Fig 9).

*Dysuria*. Two studies [4, 11] reported the incidence of dysuria, with heterogeneity between studies (p = 0.03, $I^2 = 80\%$), and the random-effect model was used for analysis. No statistically

**Table 2. Characteristics of the adverse reactions.**

| Author | EG | | | | | CG | | | | | Total |
|---|---|---|---|---|---|---|---|---|---|---|---|
| | Nausea | Dizziness | Constipation | Dysuria | Lethargy | Nausea | Dizziness | Constipation | Dysuria | Lethargy | |
| Zhang 2013 [4] | 12 | 3 | 18 | 1 | - | 28 | 30 | 35 | 12 | - | 34/105 |
| Yin 2013 [5] | 1 | - | 1 | - | 0 | 6 | - | 10 | - | 0 | 2/16 |
| Wu 2014 [6] | 3 | - | 6 | - | 3 | 12 | - | 13 | - | 10 | 13/37 |
| Han 2017 [7] | 25 | | | | | 42 | | | | | 25/42 |
| Chen 2018 [8] | 2 | - | 1 | - | 2 | 2 | - | 1 | - | 1 | 5/4 |
| Liu 2018 [9] | 4 | 3 | 6 | 0 | 1 | 9 | 4 | 14 | 0 | 1 | 14/28 |
| Liu 2019 [10] | 6 | 5 | 2 | - | 3 | 37 | 4 | 7 | - | 3 | 16/51 |
| Du 2020 [11] | 5 | 10 | 3 | 3 | 0 | 17 | 11 | 7 | 2 | 2 | 24/41 |

EG: experimental group; CG: control group.

significant difference was observed between the two groups (RR = 0.37, 95%CI: 0.02–7.02, p = 0.51 > 0.05, Fig 10).

*Occurrence of adverse reactions.* Adverse reactions were reported in all 8 studies, with the total frequency of adverse reactions occurring 133 times in the rectal administration group and 324 times in the oral administration group, about 2.4 times higher than that in the rectal administration group.

*KPS score.* KPS scores before and after treatment were reported in 2 studies [5, 9], with no heterogeneity between the studies. No significant difference was noted between the rectal and oral administration groups (RR = 1.04, 95%CI: 0.89–1.21, p = 0.63, Fig 11).

## Discussion

Pain is an unpleasant and emotional feeling, accompanied by substantial or potential tissue damage. It is a subjective feeling [12] and a common symptom among patients with advanced cancer. An estimated 40% of patients with early and mid-stage tumors and 90% with advanced tumors experience moderate to severe cancer pain, out of which 70% have not been effectively managed [13]. Opioids remain the primary clinical approach for pain management in patients with cancer-related pain; among them, OxyContin (Oshicontin) is widely administered. The current recommendations for OxyContin suggest an oral route of administration;

**Table 3. Characteristics of the included studies.**

| Author | Number of patients | Gender (M/W) | | Study | Interventions | | EG | | | | CG | | | |
|---|---|---|---|---|---|---|---|---|---|---|---|---|---|---|
| | EG/CG | EG | CG | | EG | CG | CR | PR | MR | NR | CR | PR | MR | NR |
| Zhang 2013 [4] | 44/44 | - | - | RCT | Rectal drug delivery20mg | Oral20mg | 37 | 6 | 1 | 0 | 21 | 15 | 7 | 1 |
| Yin 2013 [5] | 30/30 | - | - | RCT | Rectal drug delivery- | Oral- | 11 | 16 | 2 | 1 | 12 | 16 | 1 | 1 |
| Wu 2014 [6] | 30/30 | 17/13 | 18/12 | RCT | Rectal drug delivery10-20mg | Oral10-20mg | 11 | 15 | - | 4 | 12 | 15 | - | 3 |
| Han 2017 [7] | 64/64 | 38/26 | 33/31 | RCT | Rectal drug delivery10mg | Oral10mg | 60 | | | 5 | 59 | | | 4 |
| Chen 2018 [8] | 40/40 | 25/15 | 24/16 | RCT | Rectal drug delivery10-20mg | Oral10-20mg | 29 | 8 | - | 3 | 24 | 6 | - | 10 |
| Liu 2018 [9] | 34/34 | 23/11 | 24/10 | RCT | Rectal drug delivery10-20mg | Oral10-20mg | 18 | 11 | - | 5 | 17 | 13 | - | 4 |
| Liu 2019 [10] | 42/42 | - | - | RCT | Rectal drug delivery10mg | Oral10mg | 21 | 18 | - | 3 | 23 | 17 | - | 2 |
| Du 2020 [11] | 40/40 | 30/10 | 28/12 | RCT | Rectal drug delivery10mg | Oral10mg | 37 | | | 18 | 22 | | | 3 |

EG: experimental group; CG: control group.

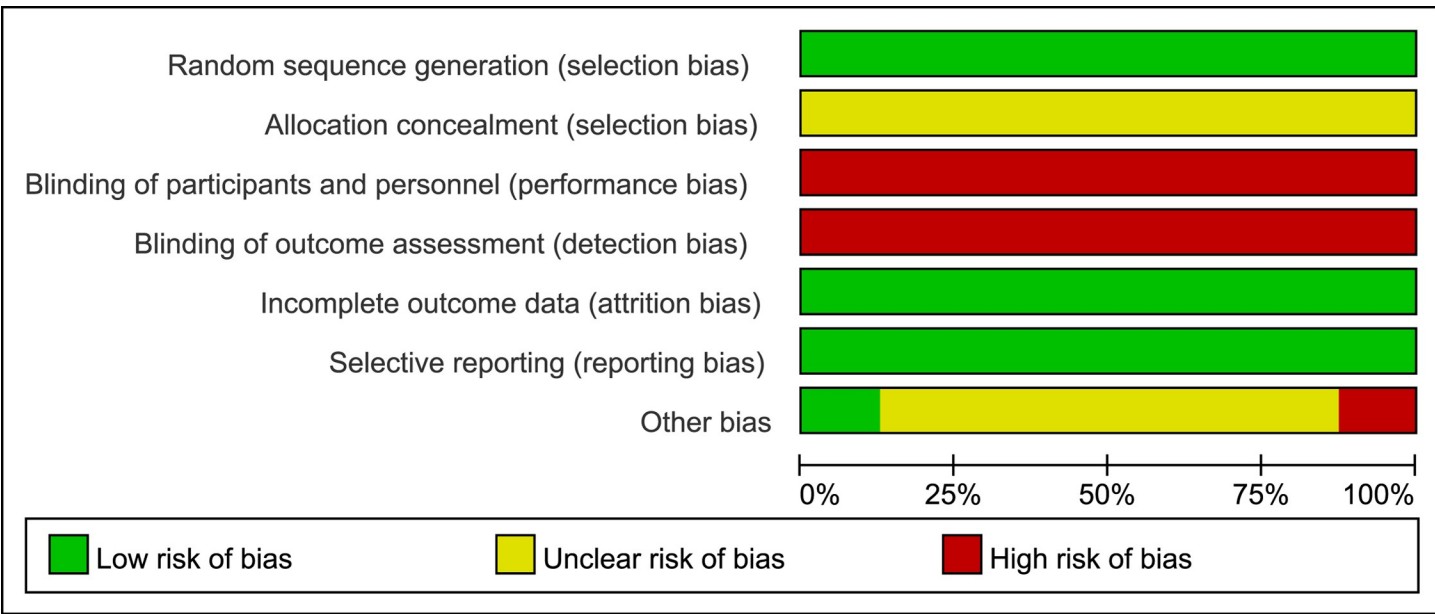

**Fig 2. Graph of bias risk ratio.**

nevertheless, it cannot be administered orally in a few patients including those with esophageal cancer having difficulties in eating. Thus, clinicians opt for rectal administration to treat pain among these patients. Moreover, unlike oral administration, rectal administration has other benefits. For example, Li Fei et al. [14] found that rectal administration minimizes the first-pass effect of the liver, prevents gastrointestinal reactions, and has a fast onset as well as enhanced safety. A total of 8 RCT studies were included in this systematic review. We found that rectal administration of OxyContin has a satisfactory clinical effect in the treatment of moderate to severe pain, with a low incidence of adverse reactions. Chen Weixian revealed that the NRS score of the observation group was better than that of the control group at 1 hour and 3 hours time points of administration, with a statistically significant difference. After 3hours, no difference was noted in the NRS score of the two groups at the statistical node. Wu Hanbing and Liu Min found no statistically significant difference in the NRS scores and pain relief efficacy between the two groups at different time points after medication. Liu Haibo and Yin Weijun noted no significant difference in the improvement of the KPS score (functional status score) between the two groups. Although treatment of cancer pain should follow oral administration principles recommended by the National Comprehensive Cancer Network (NCCN) guidelines, other administration routes may be considered for patients with eating difficulties or gastrointestinal dysfunction. For example, the use of controlled-release oral oxycodone hydrochloride (OxyContin) as a suppository has several benefits. First, it prevents the effect of nausea and vomiting caused by the primary disease on drug absorption. On the other hand, after rectal administration, the drug does not pass through the liver, thereby preventing the first-pass effect of the liver and increasing the blood drug concentration. Besides, the drug does not pass through the stomach and small intestine hence preventing its destruction by acid, alkali, and digestive enzymes, reducing its irritation in the stomach and intestines, and significantly improving its bioavailability. Using OxyContin as a suppository is cost-effective for the patient since it eliminates caregiver apprehension of high-tech equipment and supplies necessary for intravenous, intramuscular, subcutaneous, or intrathecal administration. It also affords the patient a relatively painless means of administration. Although most patients

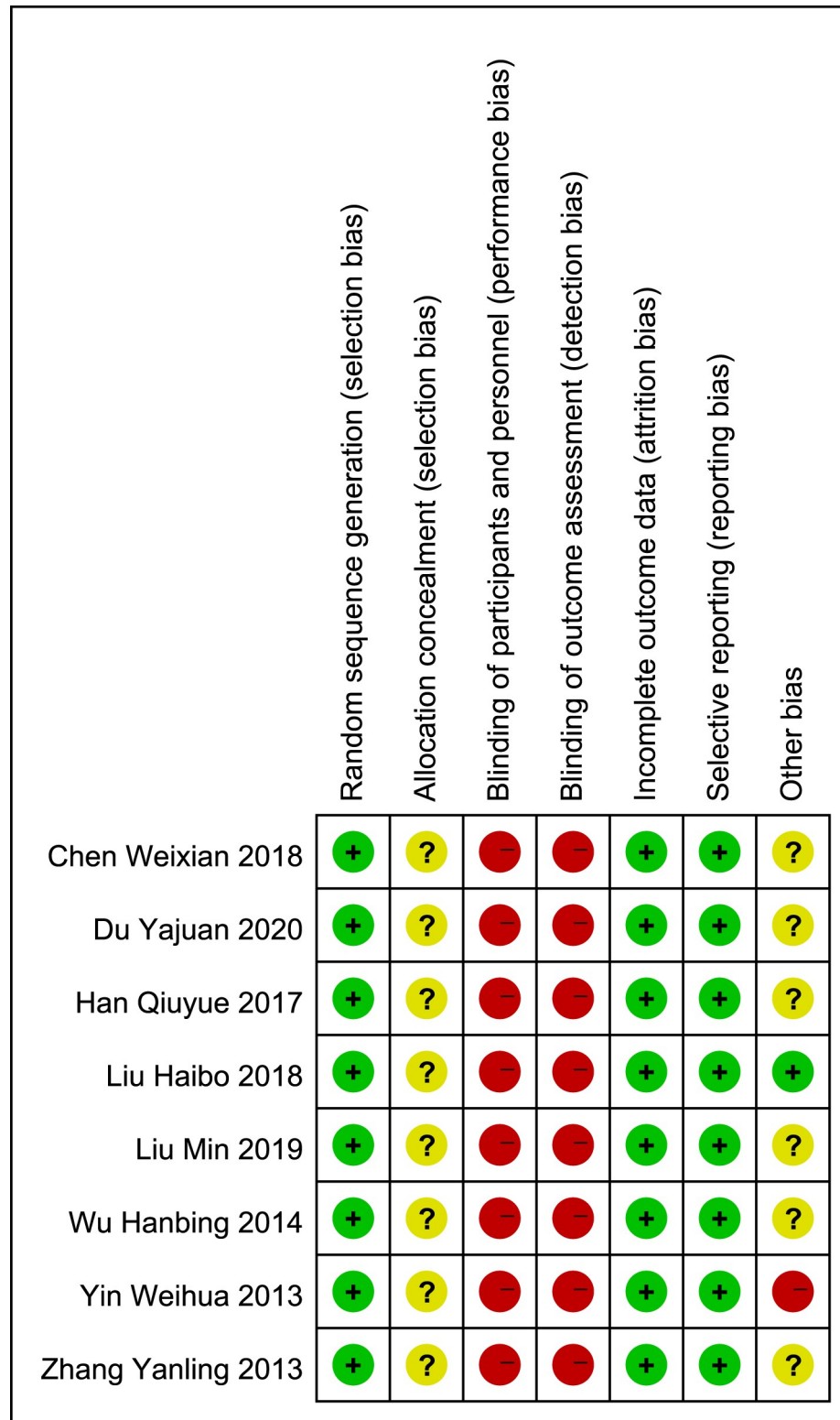

**Fig 3. Summary chart of quality evaluation.**

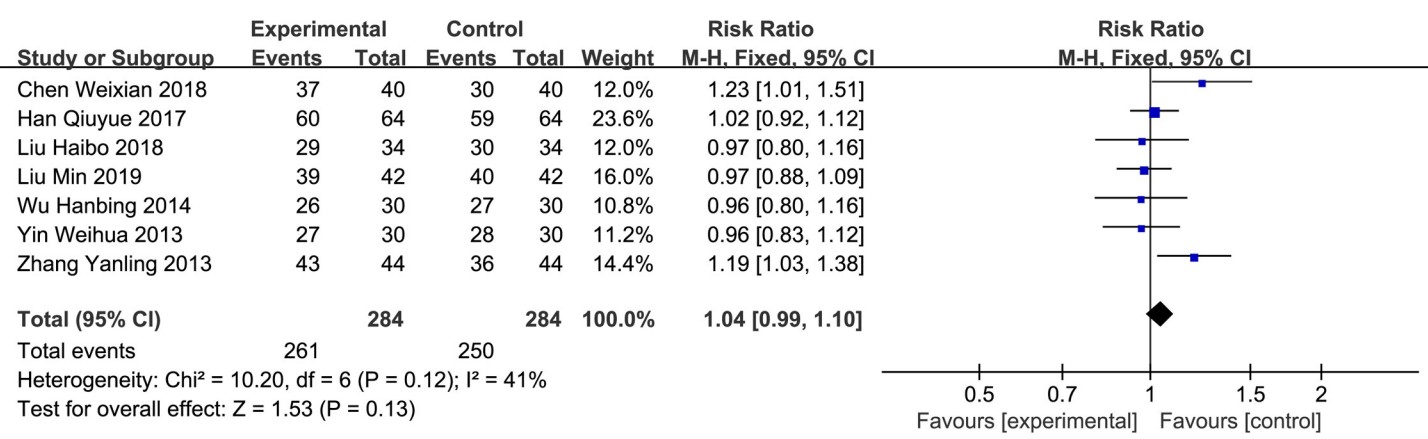

**Fig 4. Funnel plots to test for publication bias.**

manage pain using oral analgesics, simple alternatives to oral medications in the near-terminal state remain unresolved. Intramuscular, intravenous, and subcutaneous injections are uncomfortable and expensive for patients; they are also impractical in the home environment. As a rectal suppository, OxyContin effectively manages pain, facilitates the administration of

| Study or Subgroup | Experimental Events | Total | Control Events | Total | Weight | Risk Ratio M-H, Fixed, 95% CI |
|---|---|---|---|---|---|---|
| Chen Weixian 2018 | 37 | 40 | 30 | 40 | 12.0% | 1.23 [1.01, 1.51] |
| Han Qiuyue 2017 | 60 | 64 | 59 | 64 | 23.6% | 1.02 [0.92, 1.12] |
| Liu Haibo 2018 | 29 | 34 | 30 | 34 | 12.0% | 0.97 [0.80, 1.16] |
| Liu Min 2019 | 39 | 42 | 40 | 42 | 16.0% | 0.97 [0.88, 1.09] |
| Wu Hanbing 2014 | 26 | 30 | 27 | 30 | 10.8% | 0.96 [0.80, 1.16] |
| Yin Weihua 2013 | 27 | 30 | 28 | 30 | 11.2% | 0.96 [0.83, 1.12] |
| Zhang Yanling 2013 | 43 | 44 | 36 | 44 | 14.4% | 1.19 [1.03, 1.38] |
| **Total (95% CI)** | | **284** | | **284** | **100.0%** | **1.04 [0.99, 1.10]** |
| Total events | 261 | | 250 | | | |

Heterogeneity: Chi² = 10.20, df = 6 (P = 0.12); I² = 41%
Test for overall effect: Z = 1.53 (P = 0.13)

**Fig 5. Forest plot of effective rate of pain control between experimental and control group.**

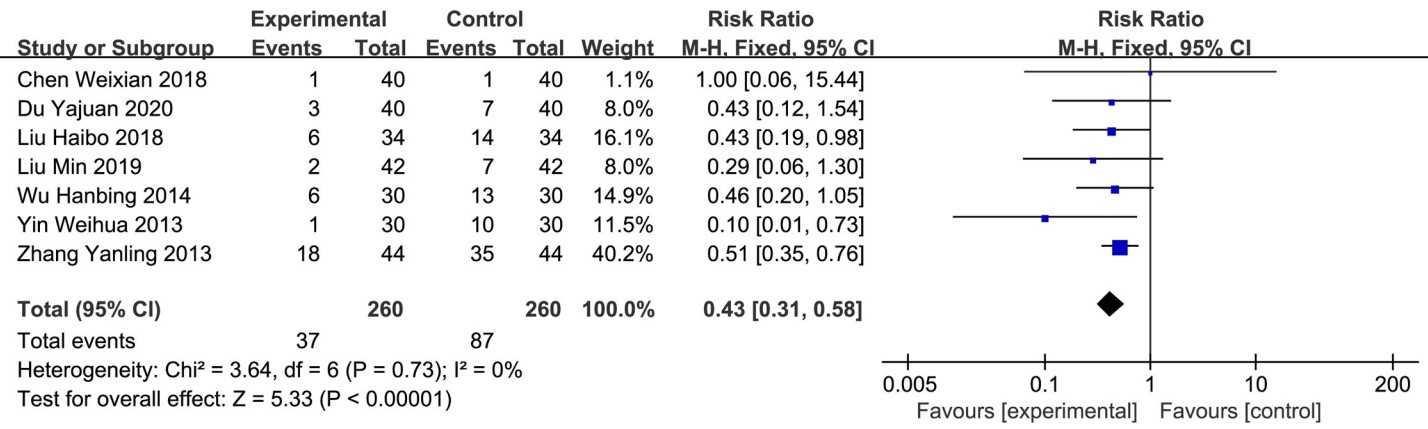

**Fig 6. Forest plot of the incidence of constipation between experimental and control group.**

caregivers, and eliminates the breakthrough pain or excessive sedation that often occurs when changing drugs. Meanwhile, animal experiments indicate that rectal administration of opioids has higher bioavailability than that of oral administration [15]. Future developments of Oxy-Contin require clinical trials investigating the bioavailability in both acute and chronic pain.

This study has compelling limitations, First, only domestic literature was included, hence lacking multi-center large sample clinical randomized controlled trial research data. Secondly, the eight included articles did not indicate allocation hiding and blinding. Thirdly, among the included literature, only Liu Haibo compared the pain control onset time and final drug dose between the two groups and found no significant difference. The remaining 7 pieces of works of literature did not indicate if there was a difference in the drug dose between the two groups during pain control. Furthermore, rectal administration has worth-mentioning limitations. First, only small doses of OxyContin are more suitable for pain control; if large doses are required, rectal administration operations will exhibit difficulties, thereby difficult acceptance by the patients. Secondly, its operation is more cumbersome than that of oral administration, and the drugs in the anus easily leak out hence requiring repeated administration.

In conclusion, the clinical application should be based on the actual situation to select the appropriate route of medication in the management of moderate to severe pain. Notably, Oxy-Contin can be considered for rectal administration. At present, rectal administration of

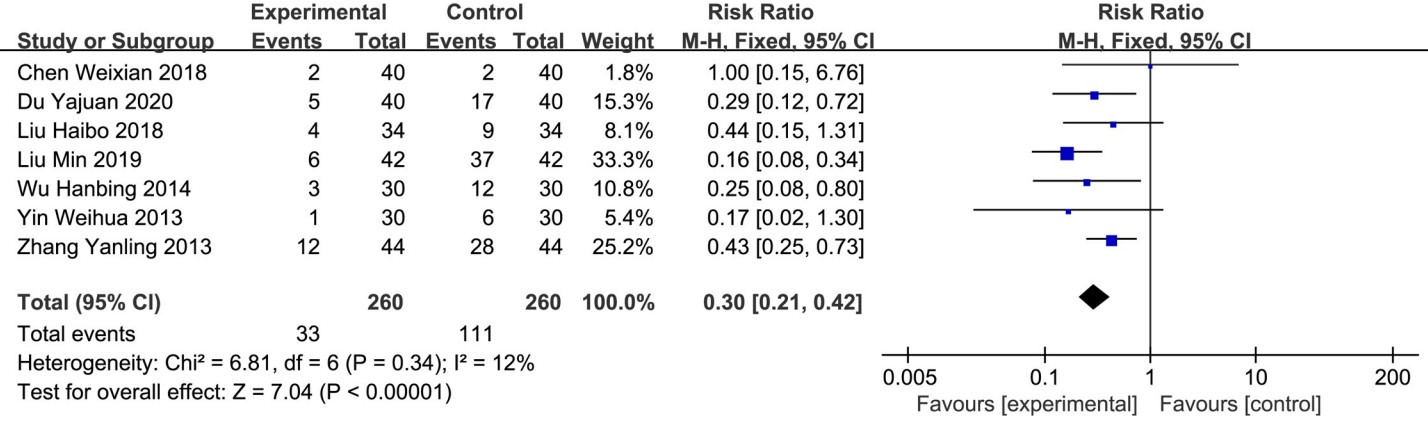

**Fig 7. Forest plot of the incidence of nausea and vomiting between experimental and control group.**

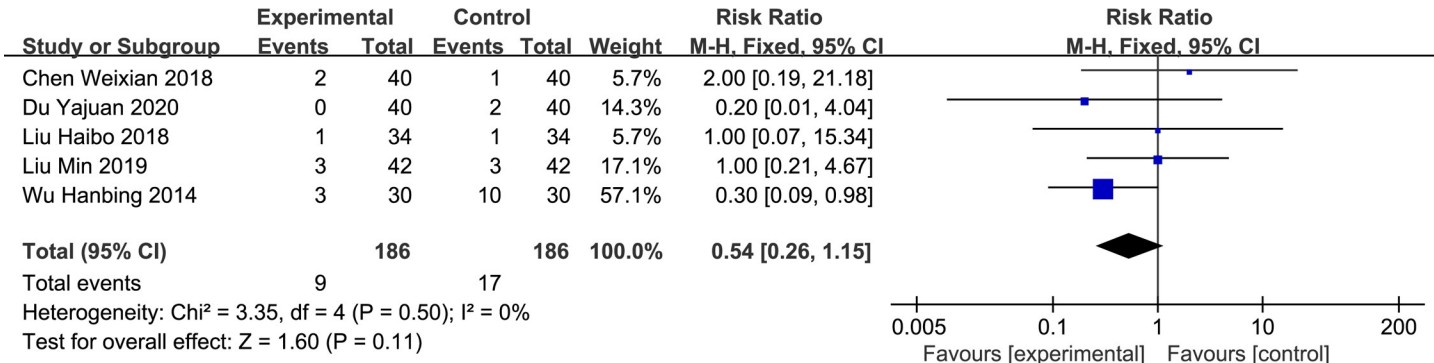

**Fig 8. Forest plot of the incidence of lethargy between experimental and control group.**

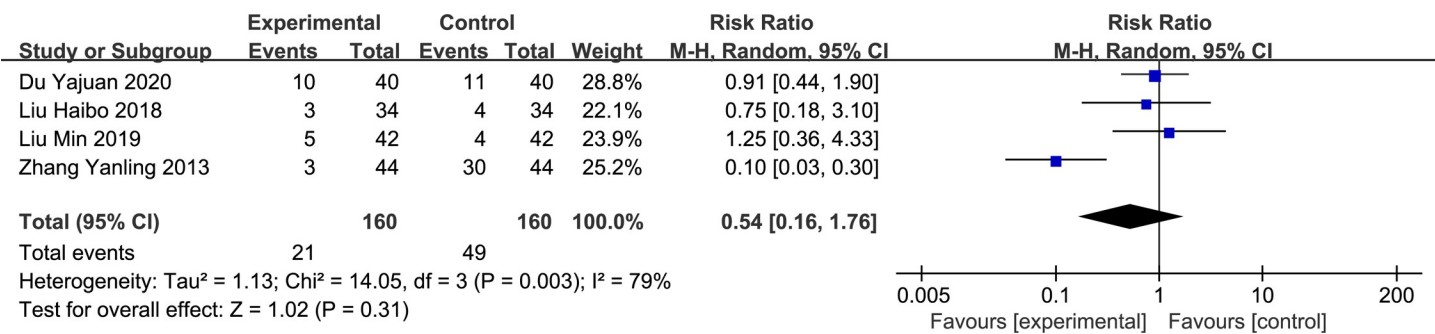

**Fig 9. Forest plot of the incidence of dizziness between experimental and control group.**

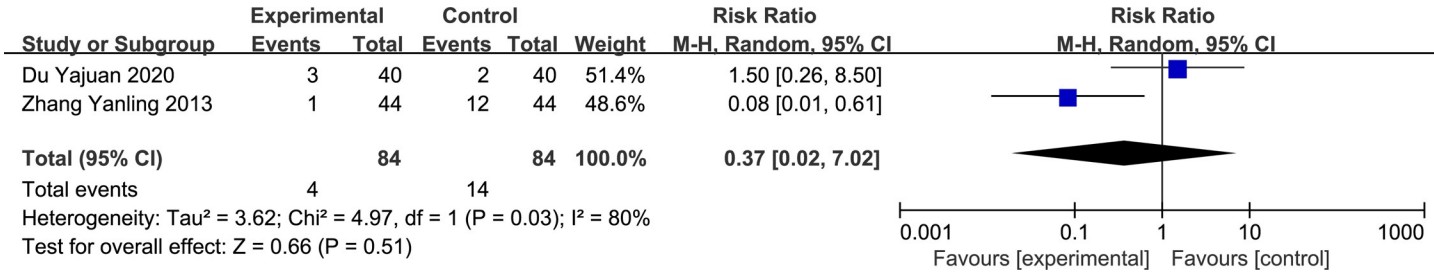

**Fig 10. Forest plot of the incidence of dysuria between experimental and control group.**

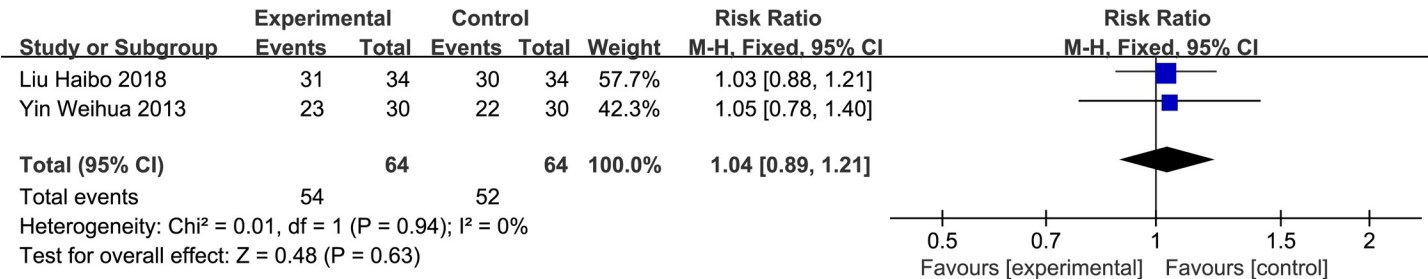

**Fig 11. Forest plot of the KPS scores between experimental and control group.**

opioids in clinical practice has not been confirmed; rectal OxyContin administration of is off-label drug use. Therefore, future studies should conduct more in-depth clinical and experimental research on the rectal administration of OxyContin. Besides, we believe that special opioids will be discovered for transrectal administration.

## Supporting information

**S1 Checklist. PRISMA checklist.**
(DOC)

## Author Contributions

**Conceptualization:** Xian Bing Hou, Dan Dan Chen.

**Data curation:** Tong Fei Cheng, Dan Wang.

**Investigation:** Xiao Jun Dai, Yao Wang.

**Methodology:** Bi Xian Cui, Yuan Yuan Wang.

**Supervision:** Hui Xu, Hong Zhou Chen.

**Writing – original draft:** Xian Bing Hou, Dan Dan Chen.

**Writing – review & editing:** Xian Bing Hou, Dan Dan Chen.

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
