## [Decision Letter · Decision Letter 0]

26 Dec 2021

PONE-D-21-30242Meta-analysis of efficacy and safety of sustained release oxycodone hydrochloride rectal administration for moderate to severe painPLOS ONE

Dear Dr. hou,

Thank you for submitting your manuscript to PLOS ONE. After careful consideration, we feel that it has merit but does not fully meet PLOS ONE’s publication criteria as it currently stands. Therefore, we invite you to submit a revised version of the manuscript that addresses the points raised during the review process.

Based on the reviewer comments, the manuscript required major revision before consider for publication. 

We look forward to receiving your revised manuscript.

Kind regards,

Girish Sailor

Academic Editor

PLOS ONE

Journal Requirements:

This work was supported by the Science and Technology Project of Fenghua Science and Technology Bureau (NO: 20186515 ).

The funders had no influence on the study design, data collection and analysis, as well as the right to publish.

Reviewers' comments:

Reviewer's Responses to Questions

**Comments to the Author**

1. Is the manuscript technically sound, and do the data support the conclusions?

Reviewer #1: Yes

Reviewer #2: Yes

2. Has the statistical analysis been performed appropriately and rigorously? 

Reviewer #1: Yes

Reviewer #2: I Don't Know

3. Have the authors made all data underlying the findings in their manuscript fully available?

Reviewer #1: Yes

Reviewer #2: Yes

4. Is the manuscript presented in an intelligible fashion and written in standard English?

Reviewer #1: No

Reviewer #2: Yes

5. Review Comments to the Author

Reviewer #1: The article is worthwhile as it addresses important clinical issues of pain management. In my professional opinion this paper will benefit the current clinical practice, however, I feel significant issues need to be seriously addressed. One of my main concern is the definition of moderate to severe pain that is not included. As this is part of the title, author must clearly define the term as per defined in all selected papers included in the review.

Initial paragraph in the discussion is repetitive to the information and phrases included in the introduction. Please revise the paragraph and assure that the there is no overlapping between these sections. The utilisation of short form for 1h and 3h that represent 'hour' should be avoided and written in full.

The paper contains quite significant grammatical errors and sentences are not written in proper English, therefore, it is suggestible for the author to send the paper to be edited by professional native English editor. This is imperative as the paper contains information that is useful for reference in clinical practice.

Eg; 'On the one hand, it can avoid the influence of nausea and vomiting caused by the primary disease on drug absorption; on the other hand, after rectal administration, the drug does not pass through the liver, thereby avoiding the first pass effect of the liver and increasing the blood drug concentration; the drug does not pass through the stomach And the small intestine, avoid the influence and destruction of acid, alkali and digestive enzymes on the medicine, reduce the irritation of the medicine to the stomach and intestines, and greatly improve the bioavailability of the medicine.'

The information that is not properly written in professional English may impede future reference and citation, hence, affecting the quality of this paper. I believe if the paper can be improved to assure that it is publishable.

Reviewer #2: The authors has produce an excellent data for this study. It is a very good alternative way to administered the mentioned medicine especially for cancer patients with difficulty to swallow. This method also will reduce several unwanted effects.

The authors has thoroughly stated the methods. There are several minor formatting issues that need to be reviewed before being accepted.

6. PLOS authors have the option to publish the peer review history of their article (what does this mean?). If published, this will include your full peer review and any attached files.

Reviewer #1: **Yes: **Norsham Juliana

Reviewer #2: No

---

## [Author Response · Author response to Decision Letter 0]

2 Mar 2022

Reviewer #1: The article is worthwhile as it addresses important clinical issues of pain management. In my professional opinion this paper will benefit the current clinical practice, however, I feel significant issues need to be seriously addressed. One of my main concern is the definition of moderate to severe pain that is not included. As this is part of the title, author must clearly define the term as per defined in all selected papers included in the review.

Answer：We would like to thank you very much for your recognition of our work and valuable comments. We've added definitions for mild, moderate, and severe pain to the Introduction.

Initial paragraph in the discussion is repetitive to the information and phrases included in the introduction. Please revise the paragraph and assure that the there is no overlapping between these sections. The utilisation of short form for 1h and 3h that represent 'hour' should be avoided and written in full.

Answer：We have truncated the repetition of the description in the introduction, modified '1h and 3h' and have written in full.

The paper contains quite significant grammatical errors and sentences are not written in proper English, therefore, it is suggestible for the author to send the paper to be edited by professional native English editor. This is imperative as the paper contains information that is useful for reference in clinical practice.

Eg; 'On the one hand, it can avoid the influence of nausea and vomiting caused by the primary disease on drug absorption; on the other hand, after rectal administration, the drug does not pass through the liver, thereby avoiding the first pass effect of the liver and increasing the blood drug concentration; the drug does not pass through the stomach And the small intestine, avoid the influence and destruction of acid, alkali and digestive enzymes on the medicine, reduce the irritation of the medicine to the stomach and intestines, and greatly improve the bioavailability of the medicine.'

The information that is not properly written in professional English may impede future reference and citation, hence, affecting the quality of this paper. I believe if the paper can be improved to assure that it is publishable.

Answer：We have already send the paper to be edited by professional native English editor. The article has corrected grammar and other errors in the original text. 

Reviewer #2: The authors has produce an excellent data for this study. It is a very good alternative way to administered the mentioned medicine especially for cancer patients with difficulty to swallow. This method also will reduce several unwanted effects.

The authors has thoroughly stated the methods. There are several minor formatting issues that need to be reviewed before being accepted.

Answer：We thank you for your careful review of the manuscript and your constructive comments. We have modified the format of the article as required by the journal.

---

## [Decision Letter · Decision Letter 1]

28 Mar 2022

Meta-analysis of efficacy and safety of sustained release oxycodone hydrochloride rectal administration for moderate to severe pain

PONE-D-21-30242R1

Dear Dr. hou,

We’re pleased to inform you that your manuscript has been judged scientifically suitable for publication and will be formally accepted for publication once it meets all outstanding technical requirements.

Kind regards,

Girish Sailor

Academic Editor

PLOS ONE

Additional Editor Comments (optional):

Reviewers' comments:

Reviewer's Responses to Questions

**Comments to the Author**

1. If the authors have adequately addressed your comments raised in a previous round of review and you feel that this manuscript is now acceptable for publication, you may indicate that here to bypass the “Comments to the Author” section, enter your conflict of interest statement in the “Confidential to Editor” section, and submit your "Accept" recommendation.

Reviewer #1: All comments have been addressed

2. Is the manuscript technically sound, and do the data support the conclusions?

Reviewer #1: Yes

3. Has the statistical analysis been performed appropriately and rigorously? 

Reviewer #1: Yes

4. Have the authors made all data underlying the findings in their manuscript fully available?

Reviewer #1: Yes

5. Is the manuscript presented in an intelligible fashion and written in standard English?

Reviewer #1: Yes

6. Review Comments to the Author

Reviewer #1: The author has addressed all comments and the manuscript is ready for target audience. The article has been rewritten with proper grammar and easy to read.

7. PLOS authors have the option to publish the peer review history of their article (what does this mean?). If published, this will include your full peer review and any attached files.

Reviewer #1: **Yes: **Norsham Juliana

---

## [Editor Report · Acceptance letter]

1 Apr 2022

PONE-D-21-30242R1 

Meta-analysis of efficacy and safety of sustained release oxycodone hydrochloride rectal administration for moderate to severe pain 

Dear Dr. Hou:

I'm pleased to inform you that your manuscript has been deemed suitable for publication in PLOS ONE. Congratulations! Your manuscript is now with our production department. 

Kind regards, 

on behalf of

Dr. Girish Sailor 

Academic Editor

PLOS ONE